# Methamphetamine and Designer Stimulants Modulate Tonic Human Cerebrovascular Smooth Muscle Contractility: Relevance to Drug-Induced Neurovascular Stress

**Nicole Hall** [1,2,†] , **Nhi Dao** [1,3,†] , **Cameron Hewett** [1] , **Sara Oberle** [1,3] , **Andrew Minagar** [1,3] , **Kariann Lamon** [1,2] , **Carey Ford** [4] , **Bruce E. Blough** [5] , **J. Steven Alexander** [1,4,*] and **Kevin S. Murnane** [1,2,6,*]

1 Louisiana Addiction Research Center, LSU Health Sciences Center at Shreveport, Shreveport, LA 71103, USA
2 Department of Pharmacology, Toxicology & Neuroscience, LSU Health Sciences Center at Shreveport, Shreveport, LA 71103, USA
3 Caddo Parish Magnet High School, Shreveport, LA 71101, USA
4 Department of Molecular & Cellular Physiology, LSU Health Sciences Center at Shreveport, Shreveport, LA 71103, USA
5 Center for Drug Discovery, Research Triangle Institute, Research Triangle Park, NC 27709, USA
6 Department of Psychiatry & Behavioral Medicine, LSU Health Sciences Center at Shreveport, Shreveport, LA 71103, USA
* Correspondence: jonathan.alexander@lsuhs.edu (J.S.A.); kevin.murnane@lsuhs.edu (K.S.M.)
† These authors contributed equally to this work.

**Abstract:** To avoid criminal prosecution, clandestine chemists produce designer stimulants that mimic the pharmacological and psychoactive effects of conventional stimulants, such as methamphetamine. Following persistent or high-dose exposure, both acute vasoconstriction and loss of vascular homeostasis are reported dangers of conventional stimulants, and designer stimulants may pose even greater dangers. To compare the effects of a conventional stimulant and two designer stimulants on vascular contraction, this study examined the direct effects of 1,3-benzodioxolylbutanamine (BDB) and N-butylpentylone in comparison to methamphetamine on the function of human brain vascular smooth muscle cells (HBVSMCs). HBVSMCs suspended in collagen gels were exposed to varying concentrations of each drug, and the degree of constriction was assessed over one week. The MTT assay was used to measure the impact of the three drugs on the cellular metabolic activity as a marker of cellular toxicity. The highest concentration tested of either methamphetamine or N-butylpentylone produced a loss of HBVSMC contractility and impaired cellular metabolism. BDB showed a similar pattern of effects, but, uniquely, it also induced vasoconstrictive effects at substantially lower concentrations. Each drug produced direct effects on HBVSMC contraction that may be a mechanism by which the cardiovascular system is damaged following high-dose or persistent exposure, and this could be exacerbated by any sympathomimetic effects of these compounds in whole organisms. BDB appears to impact HBVSMC function in ways distinct from methamphetamine and N-butylpentylone, which may present unique dangers.

**Keywords:** synthetic cathinones; methamphetamine; designer stimulant; human brain vascular smooth muscle cells; vasodilation; vasoconstriction

## 1. Introduction

There is an emerging epidemic-level rise in methamphetamine misuse in the United States. For example, reports documenting cases of methamphetamine at United States poison control centers more than quadrupled from 2000–2019 [1]. Furthermore, the Centers for Disease Control and Prevention reported that methamphetamine is a leading drug present in the victims of overdose deaths [2]. Additionally, we and many others have documented that methamphetamine damages the brain, the vasculature, and the heart, as well as other vital organs [3–9]. Therefore, this exponential growth in methamphetamine

misuse may lead to an ever-growing population of individuals with significant disease burdens that impact multiple organ systems.

Psychomotor stimulants encompass a broad range of substances with diverse chemical structures and pharmacological effects. Many misuse-related effects of methamphetamine and related stimulants are mediated by monoamine dopamine (DA) systems that are components of brain "reward" circuits (for reviews, see [10–12]). Stimulants distribute to areas of the brain rich in DA [13,14], and the visual analog scale (VAS) self-reports of drug-induced feelings of "high" correlate with the time-course of striatal stimulant distribution [15]. Stimulants often also increase levels of norepinephrine [16,17], which can contribute to their sympathomimetic effects, including profound elevation in cardiac output and systemic hypertension. Designer stimulants act through the same monoaminergic system as conventional stimulants, such as methamphetamine; however, there are reports that they can have distinct effects, different pharmacokinetic profiles, or activity at additional targets [18–20]. For example, previous studies suggest that at least some designer stimulants act at distinct pharmacological sites, including direct effects at receptors, compared to conventional stimulants [16,21–26], which could contribute to their propensities to be misused and their potential to induce toxic psychosis, hypertension, or other deleterious effects. Their capacity to induce hypertension is one of the most acutely dangerous effects of these drugs, and it may contribute to their potential to induce long-term cardiovascular damage. While their acute sympathomimetic effects are believed to play a major role in these acute and persistent cardiovascular risks, mechanisms such as direct effects on physiologically relevant cardiovascular cells may also contribute.

Designer stimulants, including novel amphetamines as well as their β-ketone analogs, the synthetic cathinones, represent one of the most prevalent, widely misused, and dangerous classes of novel psychoactive substances (NPSs). NPSs are chemically similar to classic psychoactive substances (e.g., stimulants, hallucinogens, and cannabinoids) such that they produce similar psychoactive effects, but they are dissimilar enough to evade legal prosecution. With the classification of an NPS, additional NPSs with slight variations in chemical structure are synthesized to meet the market demand. The United Nations Office on Drugs and Crime (UNODC) and the European Monitoring Centre for Drugs and Drug Addiction (EMCDDA) have been tracking the rise in NPS use. Often, NPS are marketed as "not for human consumption" and sold as "bath salts" or "research chemicals." While many of these compounds have been scheduled into the most restricted drug classes, this process can be time-consuming and numerous designer stimulants continue to emerge, oftentimes faster than they can be controlled.

It is often the case that very little data exist about the pharmacological properties, metabolism, and toxicity of designer stimulants. Given the lengthy process to control their distribution, it would be useful if there were a rapid method for assessing the potential cardiovascular toxicity of these agents. Moreover, there are reports that methamphetamine can damage the cardiovascular system through mechanisms other than sympathomimetic effects. Vascular smooth muscle cells encase the endothelium, which enables interactions with endothelial cells to regulate and maintain vascular function [27]. Recent reports have indicated that methamphetamine induces apoptosis in vascular smooth muscle cells [28] and remodeling of pulmonary arteries [29]. Evidence from methamphetamine users also points to impaired smooth muscle vasodilatory function [30]. In the present study, we therefore extended this research by determining the effects of methamphetamine on human brain vascular smooth cells (HBVSMC) using a novel high-throughput assay that could be engineered to rapidly assess the potential of emerging designer stimulants to induce cardiovascular damage. The effects of methamphetamine were compared to the designer amphetamine analogs 1,3-benzodioxolylbutanamine (BDB) and n-butylpentylone. These analogs were chosen because they contain methylendioxy moieties likely to increase their capacities to induce serotonin release. Serotonin release is known to contribute to cardiovascular damage through activity at serotonin 2B receptors, and recent research suggests that serotonin release may augment toxicity at vascular smooth muscle cells [31].

We hypothesized that methamphetamine would cause direct damage at HBVSMCs and that BDB and n-butylpentylone would exhibit even greater disruption of vascular function than methamphetamine.

## 2. Materials and Methods

### 2.1. Cell Culture

HBVSMC (ScienCell, Carlsbad, CA, USA) were cultured in smooth muscle cell medium (SMCM, ScienCell) supplemented with 0.04% doxycycline (Sigma, St. Louis, MO, USA), 2% fetal bovine serum (FBS, ScienCell), 1% smooth muscle cell growth supplement (SMCGS, ScienCell), and 1% penicillin/streptomycin solution (P/S ScienCell). To facilitate cell attachment, T-75 flasks were coated or pre-treated with poly-L-lysine. HBVSMC were seeded on flasks containing 10 mL of fortified SMCM and were incubated at 37 °C with 5% $CO_2$. Media were changed every 2–3 days until ~90% confluency was reached.

### 2.2. Rat-Tail Type-1 Collagen Preparation

Rat tail type-1 collagen was prepared by modifying the protocol previously described [32,33]. Briefly, rat tail tendons were extracted, washed in 70% ethanol (Sigma), and dissolved in 2 mM acetic acid (Sigma) for one week at 4 °C under constant agitation. Collagen was filtered through a 200 μm nylon filter, aliquoted, snap-frozen, and freeze-dried using a bench-top manifold freeze-dryer (Millrock Technology, Kingston, NY, USA). Aliquots were stored at −20 °C until ready for use.

### 2.3. Preparation of HBVSMC/Collagen Gels

For the collagen gel contraction assay, HBVSMC/collagen gels were prepared with modifications to the procedure described previously [33]. Prior to experiments, freeze-dried collagen was resolubilized at a final concentration of 1.25 mg/mL in cold 0.012 M hydrochloric acid under gentle agitation at 4 °C overnight. On the day of the experiment, cultured HBVSMC were harvested using trypsin/EDTA. The HBVSMC were centrifuged at 1600 rpm for 5 min, suspended in SMCM and counted using an automated cell counter (EVE, NanoEnTek, Seoul, South Korea). To attain 50,000 cells per well, $1.5 \times 10^6$ cells were resuspended in 10 mL of supplemented SMCM, into which 4 mL of the resolubilized collagen gel solution, 1 mL of cold 5X PBS, and 6.25 μL of 1 M NaOH were added and mixed by vortex. The final HBVSMC/collagen gel mixture (10 mL of cell suspension in 4 mL of collagen gel solution) was seeded on a 24-well plate by aliquoting 500 μL of the mixture into each well. The plate was incubated at 37 °C for 1 h for polymerization.

### 2.4. Drugs and Chemicals

BDB and n-butylpentylone were provided by Dr. Blough of the Research Triangle Institute (Research Triangle Park, NC, USA). Methamphetamine HCl was obtained from a commercial vendor (Cayman Chemical Company, Ann Arbor, MI, USA). The chemical structures are detailed in Figure 1. All drug concentrations reported were calculated from the salt weight and dissolved in saline.

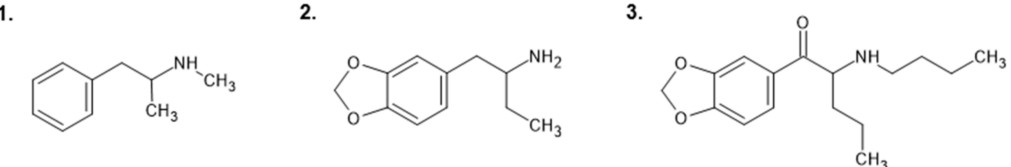

**Figure 1.** Chemical structures of (1) Methamphetamine, (2) 1,3-Benzodioxolylbutanamine (BDB), and (3) N-butylpentylone.

### 2.5. Exposure Regimen

Stock solutions of $2 \times 10^{-3}$ M for each of methamphetamine, BDB, and n-butylpentylone were prepared in supplemented SMCM and sterilized by syringe filter (Santa Cruz Biotech-

nology). Serial dilutions of $2 \times 10^{-4}$ M, $2 \times 10^{-5}$ M, $2 \times 10^{-6}$ M and $2 \times 10^{-7}$ M were prepared from stock solutions. One 24-well plate was prepared for each drug tested. A total of 500 µL of supplemented SMCM was added to each well of a four-well column as control. One of the five dilutions prepared was added 500 µL per well to each four-well column remaining, bringing the final concentrations of the treatments to $10^{-3}$ M–$10^{-7}$ M. A wide range of concentrations were chosen that envelop the physiological concentrations of methamphetamine but allow for potency differences that could be present in the lesser studied designer stimulants. The cells were treated once and placed in a $CO_2$ incubator at 37 °C for one week. The plates were removed daily for image collection and immediately returned to the incubator.

### 2.6. Image Collection and Analysis

The plates were photographed daily to track the contraction of the gels. Representative images of the gels after one day of treatment are shown in Figure 2. To photograph the wells, a black box was crafted to hold the plates. Mirrors were organized to reflect the bottom of the plate for optimal visualization of the gels. To account for curvature in the outer edges of the frame, three photos of each plate (top, center, bottom) were taken daily. For each well, the areas of the well and of the gel occupying the well were each determined using ImageJ (NIH). The percentage of well occupied was calculated by dividing the area of the gel by the area of the well. The contraction was calculated by subtracting the percentage of well occupied from 100. The data were normalized by dividing the percent contraction of the well by the average percent contraction of the control wells.

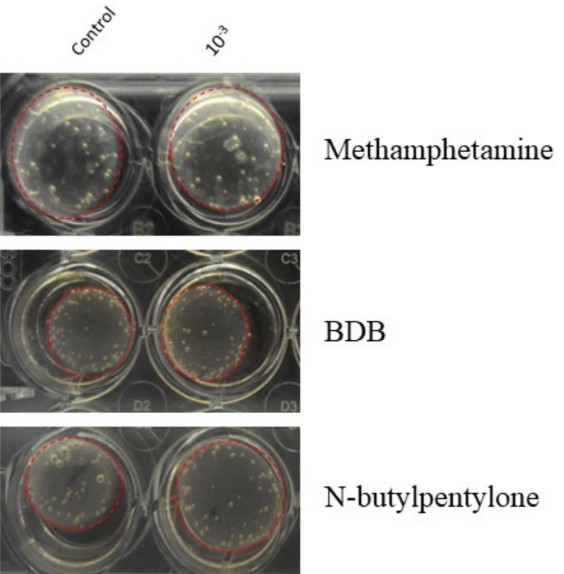

**Figure 2.** Representative images of collagen gels seeded with human brain vascular smooth muscle cells (HBVSMC) in 24-well plates after one day of treatment. Dashed red circles outline the gels within the well. Control wells on the left contain supplemented smooth muscle cell media (SMCM). Wells on the right are treated with $10^{-3}$ M concentrations of each drug tested (top to bottom): Methamphetamine, 1,3-Benzodioxolylbutanamine (BDB), and N-butylpentylone.

### 2.7. MTT Assay

The MTT assay was performed as an indicator of cell viability. Stock solutions of $10^{-3}$ M for methamphetamine, BDB, and n-butylpentylone were each prepared in supplemented SMCM and sterilized by syringe filter (Santa Cruz Biotechnology, Dallas, TX, USA). Serial dilutions of $10^{-4}$ M, $10^{-5}$ M, $10^{-6}$ M and $10^{-7}$ M were prepared from stock solutions. HBVSMC were seeded at 10,000 cells per well in a 96-well tissue culture-treated plate and incubated at 37 °C overnight for attachment. Cells were treated by adding 100 µL of SMCM for control or each of the five drug dilutions for testing in triplicate. After 24 h, the media

were aspirated, and 100 μL of phenol-red-free DMEM and 10 μL of sterile filtered MTT solution (5 mg/mL, dissolved in PBS) were added to each well. The plate was incubated at 37 °C for 3 h. The media were aspirated, and 100 μL of DMSO was pipetted up and down to mix. After 15 min on a rocker at room temperature, absorbance was measured using a spectrophotometer at 590 nm with a reference filter of 650 nm.

*2.8. Data Analysis*

Graphs were compiled using GraphPad Prism (GraphPad Software Inc.; La Jolla, CA, USA). Results are presented as the mean ± standard error of the mean (SEM). Statistical analysis was performed by one-way analysis of variance (ANOVA), and unequal variance was determined with a Dunnett's post hoc test. Outliers were identified by Grubb's Test and eliminated. Statistical significance is represented by a single asterisk for $p < 0.05$, double asterisks for $p < 0.01$, and triple asterisks for $p < 0.001$.

## 3. Results

*3.1. Effects of Methamphetamine on HBVSMC*

We examined the effects of methamphetamine on HBVSMCs using the collagen gel assay (Figure 3A). There was a significant main effect ($F_{(5,18)}$ = 11.09; $p < 0.001$) of treatment at four days of exposure to methamphetamine. Post hoc analysis revealed a 50% reduction in contraction ($p < 0.001$) between the $10^{-3}$ M treatment and control (Figure 3B). Additionally, there was a significant main effect ($F_{(5,18)}$ = 19.40; $p < 0.001$) of treatment at seven days of treatment and a 75% reduction in contraction ($p < 0.001$) between the $10^{-3}$ M treatment and control (Figure 3C).

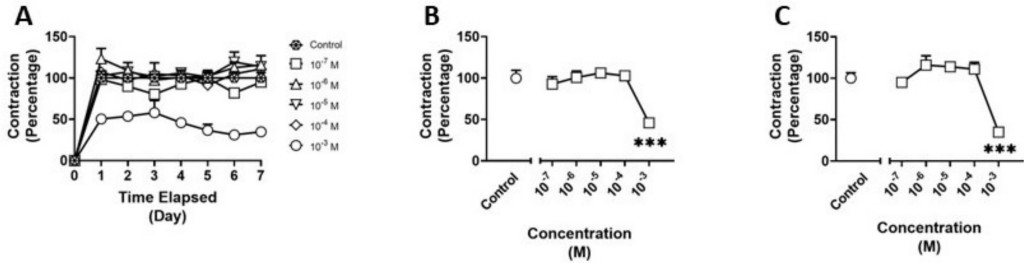

**Figure 3.** Contraction of human brain vascular smooth muscle cells (HBVSMC) in collagen gel assay (**A**) after treatment with varying concentrations of methamphetamine for one week, (**B**) on day 4 of treatment, and (**C**) on day 7 of treatment. Data are normalized to negative controls. Statistical analysis was performed using one-way ANOVA with post hoc analysis by Dunnett's multiple comparisons test. Error bars represent ± SEM. Asterisks indicate statistical significance compared to non-treated controls: *** = $p < 0.001$.

*3.2. Effects of BDB on HBVSMC*

We examined the effects of BDB on HBVSMCs using the collagen gel assay (Figure 4A). There was a significant main effect ($F_{(5,17)}$ = 18.67; $p < 0.001$) of treatment at four days of exposure to BDB. Post hoc analysis revealed a significant reduction in contraction between control and treatment at a concentration of $10^{-3}$ M ($p < 0.001$) and a 10% increase in contraction between control and $10^{-7}$ M treatment ($p < 0.01$) (Figure 4B). Additionally, there was a significant main effect ($F_{(5,17)}$ = 5.582; $p = 0.0032$) of treatment at seven days of treatment and a significant increase in contraction ($p < 0.01$) between treatment at a concentration of $10^{-7}$ M and control at day seven of exposure (Figure 4C).

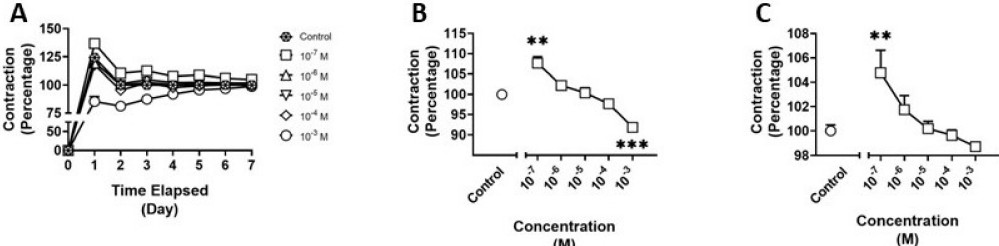

**Figure 4.** Contraction of human brain vascular smooth muscle cells (HBVSMC) in collagen gel assay (**A**) after treatment with varying concentrations of 1,3-benzodioxolylbutanamine (BDB) for one week, (**B**) on day 4 of treatment, and (**C**) on day 7 of treatment. Data are normalized to negative controls. Statistical analysis was performed using one-way ANOVA with post hoc analysis by Dunnett's multiple comparisons test. Error bars represent $\pm$ SEM. Asterisks indicate statistical significance compared to non-treated controls: ** = $p < 0.01$ and *** = $p < 0.001$.

### 3.3. Effects of N-butylpentylone on HBVSMC

We examined the effects of n-butylpentylone on HBVSMCs using the collagen gel assay (Figure 5A). There was a significant main effect ($F_{(5,18)}$ = 15.41; $p < 0.001$) of treatment on day four of exposure to n-butylpentylone. Post hoc analysis revealed a 20% reduction in contraction ($p < 0.001$) between the $10^{-3}$ M treatment and control (Figure 5B). Additionally, there was a significant main effect ($F_{(5,18)}$ = 14.48; $p < 0.001$) of treatment on day seven and a 20% reduction in contraction ($p < 0.001$) between the $10^{-3}$ M treatment and control (Figure 5C).

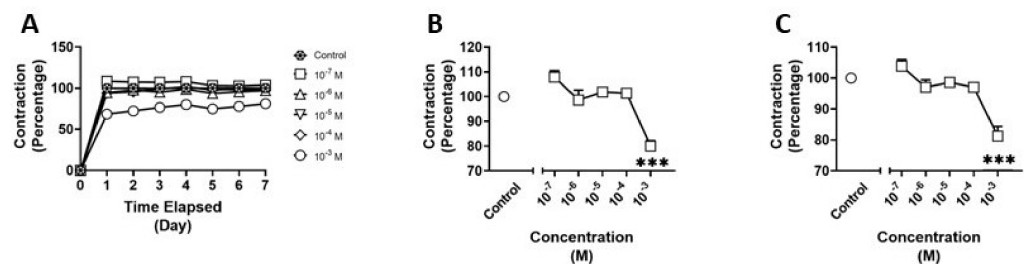

**Figure 5.** Contraction of human brain vascular smooth muscle cells (HBVSMC) in collagen gel assay (**A**) after treatment with varying concentrations of n-butylpentylone for one week, (**B**) on day 4 of treatment, and (**C**) on day 7 of treatment. Data are normalized to negative controls. Statistical analysis was performed using one-way ANOVA with post hoc analysis by Dunnett's multiple comparisons test. Error bars represent $\pm$ SEM. Asterisks indicate statistical significance compared to non-treated controls: *** = $p < 0.001$.

### 3.4. MTT Data

We evaluated the effects of methamphetamine, BDB, and n-butylpentylone on the cellular metabolic activity of HBVSMCs. There was a significant main effect ($F_{(2,21)}$ = 108.2; $p < 0.001$) of treatment after exposure to methamphetamine. Post hoc analysis revealed a significant decrease in cellular metabolic activity ($p < 0.001$) between control and treatment at a concentration of $10^{-3}$ M (Figure 6A). There was a significant main effect ($F_{(2,29)}$ = 166.6; $p < 0.001$) of treatment after exposure to BDB. Post hoc analysis revealed a significant decrease in cellular metabolic activity ($p < 0.001$) between control and treatment at a concentration of $10^{-3}$ M (Figure 6B). There was a significant main effect ($F_{(2,29)}$ = 319.5; $p < 0.001$) of treatment after exposure to n-butylpentylone. Post hoc analysis revealed a significant decrease in cellular metabolic activity ($p < 0.001$) between control and treatment at a concentration of $10^{-3}$ M (Figure 6C).

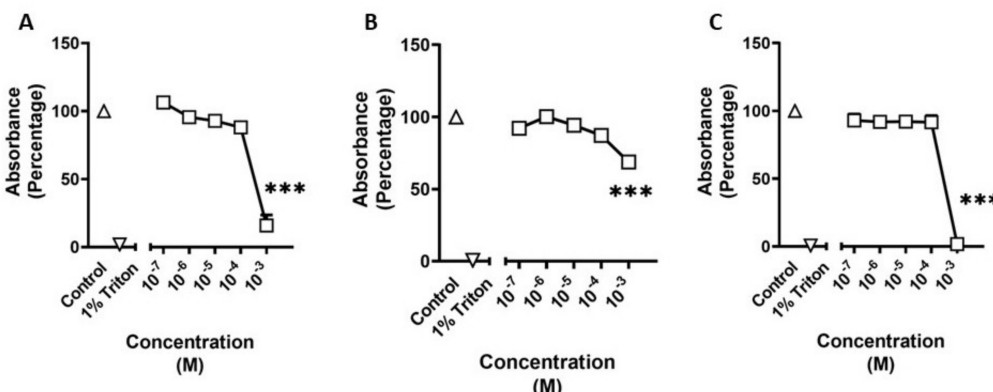

**Figure 6.** Cellular metabolic activity of human brain microvascular smooth muscle cells (HBVSMC) as determined through MTT assay after treatment with varying concentrations of methamphetamine (**A**), 1,3-benzodioxolylbutanamine (BDB) (**B**), and n-butylpentylone (**C**). Positive control is treatment with 1% Triton X-100. Data are normalized to negative controls. Statistical analysis was performed using one-way ANOVA with post hoc analysis by Dunnett's multiple comparisons test. Error bars represent ± SEM. Asterisks indicate statistical significance compared to non-treated controls: *** = $p < 0.001$.

## 4. Discussion

While it is known that methamphetamine damages the brain and vasculature, the direct effects of methamphetamine and related designer stimulants on physiologically relevant cells are understudied. In the current study, we utilize a collagen gel contraction assay to assess the impact of exposure to methamphetamine, BDB, and n-butylpentylone on HBVSMC contractile functionality. The MTT assay was used to evaluate the cytotoxicity of the compounds on the HBVSMC based on their capacity to disrupt cellular metabolic activity. We demonstrate that millimolar concentrations of each drug tested have the capacity to significantly reduce the contraction of HBVSMC and, likewise, produce a significant reduction in cellular metabolic activity, suggesting that the dilatory effects are due to cytotoxicity. Furthermore, we show that BDB causes increased contraction at low concentrations not seen with methamphetamine or n-butylpentylone treatment. Together, these data suggest that BDB is acting directly on the HBVSMC by mechanisms distinct from those of methamphetamine and n-butylpentylone.

There are few studies investigating the effects of methamphetamine on cerebral vascular tone and even fewer studying designer amphetamine analogs. One study utilizing both brain endothelial cells and cerebral arteriolar vessels isolated from mice determined that methamphetamine-induced vasoconstriction was mediated by the release of endothelin-1 [34]. Studies have shown that methamphetamine causes vascular changes that result in a sustained reduction in cerebral blood flow [35] and striatal hypoxia [36]. Though the effects of methamphetamine-induced vascular changes and vasoconstriction are investigated in these works, the effects directly on the smooth muscle cells have not been explored. Reviews studying the literature related to synthetic cathinones have exposed that methylenedioxypyrovalerone (MDPV), a β-keto analog of amphetamine, has been found to induce brain hyperthermia through an increase in peripheral vasoconstriction [20]; however, the designer stimulants used in this study, though similar in structure, were not directly investigated in the studies reported. The sparse research on BDB has focused mainly on its capacity to affect monoamine re-uptake and release [37] and its metabolism [38].

Although this study is not mechanistic in nature, the data herein suggest that BDB is acting on HBVSMC through a mechanism unlike that of methamphetamine or n-butylpentylone. Using the model in Figure 7, we illustrate the possibility of differing mechanisms for influencing vascular constriction between the drugs tested. As a sympathomimetic, methamphetamine is generally thought to cause vasoconstriction through the activation of the sympathetic nervous system. It appears BDB is acting directly on the HBVSMC to cause vasoconstriction, perhaps through the stimulation of the α1 adrener-

gic receptor. Utilizing prazosin, an α1 receptor antagonist, future studies could test this potential mechanism of action of BDB.

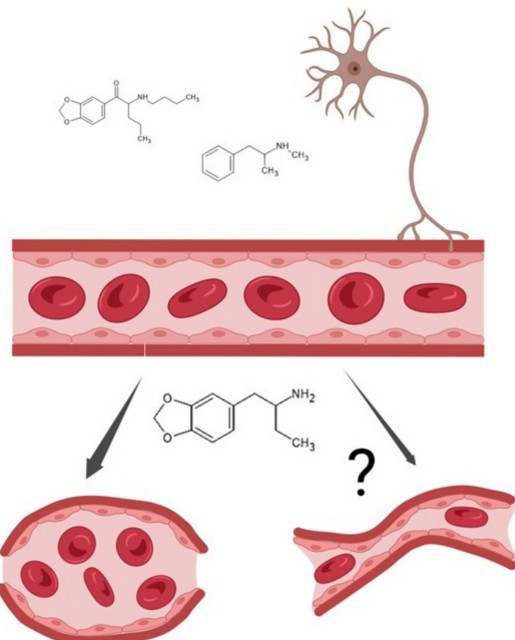

**Figure 7.** Model demonstrating possible mechanistic differences allowing for BDB to affect vascular constriction directly upon vascular smooth muscle cell activation in response to low concentrations of the drug. The data suggest BDB is acting directly on HBVSMC at low concentrations through a mechanism, unlike methamphetamine and n-butylpentylone. The former could be causing vasoconstriction by stimulating α1 adrenergic receptors, while the latter may require activation of the sympathetic nervous system for vasoconstriction to occur.

Additionally, while it is important to identify the consequences of exposure on individual cell types, the vasculature is comprised of multiple cell types, each reacting in its own way to drug exposure. Furthermore, drug-induced increases in vasodilatory compounds, such as nitric oxide, could cause opposing relaxation. The combined reactions of the vasculature as a unit and/or the presence of vasodilatory compounds may cause results dissimilar to those contained in this study. Future in vivo studies could include the implantation of telemetric probes to facilitate hypertension studies evaluating the capacity of BDB as a vasopressor to investigate potential cardiac pathophysiology resulting from drug exposure.

The present study investigates the effects of chronic exposure to the tested stimulants. While the acute effects of these drugs have clinical implications, there is, to the best of our knowledge, no evidence that these compounds directly interact with smooth muscle cells of the vasculature. As discussed above, the acute vasoconstrictive effects of methamphetamine are mediated through indirect agonism of adrenergic receptors via the release of norepinephrine. Our model, which focuses on direct effects on smooth muscle cells, does not recapitulate these complex interactions. Future studies incorporating cotreatment with norepinephrine could facilitate investigations of acute exposure.

These novel analogs could be seen as safer options compared to illicit substances, as there is a lack of information regarding potential dangers. To our knowledge, this is the first paper to explore the effects of designer stimulants on the contractility of HBVSMC. The findings reported here illustrate possible complications that could be encountered by individuals exposed to chemical analogs of commonly misused drugs. The chemical compounds selected for this study constitute a limited representation of how slight alterations to a drug's chemical structure can result in altered effects. The collagen gel assay used in this study is an effective method for efficiently screening a drug's capacity for altering

the vasoactive functionality of smooth muscle cells. Future experiments that include this technique for a large-scale structure–activity relationship study could elucidate specific structural components that impact vasomotion.

## 5. Conclusions

The effects of methamphetamine, BDB, and n-butylpentylone on the contraction and cellular metabolic activity of HBVSMC were explored. We determined that BDB alone induced the contraction of HBVSMC likely by mechanisms different from the chemically similar substances methamphetamine and n-butylpentylone. These studies highlight the potential dangers of designer drugs previously unknown and warrant further study to elucidate possible pathophysiology related to exposure.

**Author Contributions:** Conceptualization, C.H., N.D., K.S.M. and J.S.A.; methodology, N.H., N.D., S.O., A.M., K.L., C.F., K.S.M. and J.S.A.; formal analysis, N.H., N.D., K.L. and C.H.; investigation, N.D., S.O., K.L., A.M. and C.H.; resources, K.S.M., J.S.A. and B.E.B.; data curation, N.H., C.H., N.D. and K.L.; writing—original draft preparation, N.H., N.D., C.H. and K.S.M.; writing—review and editing, N.H., N.D., C.H., K.S.M. and J.S.A.; visualization, N.H. and K.S.M.; supervision, N.H., C.H., J.S.A. and K.S.M.; project administration, K.S.M. and J.S.A.; funding acquisition, K.S.M. All authors have read and agreed to the published version of the manuscript.

**Funding:** This work was supported by an Institutional Development Award from the National Institutes of General Medical Sciences of the National Institutes of Health under grant No. P20GM121307 and by T32HL155022 from NIH NHLBI, awarded to A. Wayne Orr and Karen Y. Stokes, Center for Cardiovascular Diseases and Sciences, LSU Health Sciences Center Shreveport, as well as the Louisiana Addiction Research Center.

**Institutional Review Board Statement:** Not applicable.

**Informed Consent Statement:** Not applicable.

**Data Availability Statement:** The data presented in this study are available on request from the corresponding authors.

**Conflicts of Interest:** The authors declare no conflict of interest.

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
