# Peer review of "Methamphetamine and Designer Stimulants Modulate Tonic Human Cerebrovascular Smooth Muscle Contractility: Relevance to Drug-Induced Neurovascular Stress"

_pathophysiology, doi:10.3390/pathophysiology30020013_

Round 1

Reviewer 1 Report

The manuscript by N. Hall et al., entitled “Methamphetamine and Designer Stimulants Modulate Tonic Human Cerebrovascular Smooth Muscle Contractility: Relevance to Drug Induced Neurovascular Stress” is presented for review.

In this study the Authors investigate the effects of various concentrations (10-7M-10-3M) of methamphetamine, 1,3-benzodioxolylbutanamine (BDB), and n-butylpentylone on contraction and cellular metabolic activity/cytotoxicity in human brain vascular smooth muscle cells (HBVSMC) in culture. The presented findings indicate that all compounds (particularly at higher concentration; e.g. 10-3M) are cytotoxic to HBVSMC (MTT assay) and reduce/impair HBVSMC contractility as assessed by a novel high-throughput collagen gel assay that could be engineered to rapidly assess the potential of emerging designed stimulants to induce cardiovascular damage. The findings also indicate that BDB affects HBVSMC contractility at lower concentrations as compared to methamphetamine or n-butylpentylone suggesting different mechanism(s) of action. 

It is this reviewer’s opinion that the study is very timely and, while not mechanistic in nature, it provides with a new knowledge on the subject. The experiments are straightforward, well executed, and the results are self-explanatory. Overall, the manuscript is well written and was easy to follow. Some study limitations are acknowledged and discussed.

Despite overall enthusiasm, however, there are some issues requiring further clarification: 

1) “Abstract-line 28”. “….and both constriction and dilation were assessed over one week.” The presented results deal with HBVSMC contraction only! There are no data showing “dilation/relaxation” of the HBVSMC. Please clarify.

2) “Abstract-lines 30-32”. “The highest concentration tested of either methamphetamine or N-butylpentylone produced a loss of HBVSMC homeostasis and impaired cellular metabolism”. Since your study deals specifically with HBVSMC contraction, it would be more appropriate to use term “contraction” instead of “homeostasis” in the above sentence.

3) “Abstract-line 34”. The Authors state that “Each drug produced direct effects on HBVSMC vascular homeostasis….”. Based on the presented results this statement is overreaching and would be more suitable to be included in to “Discussion” section. This study deals with HBVSMC responses in culture only, and therefore, “vascular homeostasis“ per se was not directly investigated. Please clarify. 

4) Representative images of HBVSMC contraction in the collagen gel (e.g. in the absence and presence of high concentration of the compound(s)) would help the reader with data visualization/interpretation.

5) “Results”: Consider removing repetitive statements relevant to the use of statistical analysis in data interpretation for all Figures. Statistical analysis/approaches are well-described in the “Methods” section and in the Figure legends, as well.

6) Please label individually each panel in the Figures 1-4 and use this labeling in describing your findings in the “Results” section to help the reader with data interpretation.

7) Figure 3. The data presented in Figure 3 (left panel) shows no difference in HBVSMC contraction in response to stimulation with 10-3M BDB on days 4 and 7, while the Figure 3 (center and right panels) indicate significant differences at these specific time points. Please clarify.

8) Figure 6. The Figure/Diagram requires more detailed description/explanation of the potential effects/mechanisms involved.

Author Response

1) “Abstract-line 28”. “….and both constriction and dilation were assessed over one week.” The presented results deal with HBVSMC contraction only! There are no data showing “dilation/relaxation” of the HBVSMC. Please clarify.

We removed mention of dilation for clarity. Thank you

2) “Abstract-lines 30-32”. “The highest concentration tested of either methamphetamine or N-butylpentylone produced a loss of HBVSMC homeostasis and impaired cellular metabolism”. Since your study deals specifically with HBVSMC contraction, it would be more appropriate to use term “contraction” instead of “homeostasis” in the above sentence.

The language of the abstract was corrected for clarity and references to vascular homeostasis were removed. Thank you

3) “Abstract-line 34”. The Authors state that “Each drug produced direct effects on HBVSMC vascular homeostasis….”. Based on the presented results this statement is overreaching and would be more suitable to be included in to “Discussion” section. This study deals with HBVSMC responses in culture only, and therefore, “vascular homeostasis“ per se was not directly investigated. Please clarify. 

The language of the abstract was corrected for clarity. Thank you

4) Representative images of HBVSMC contraction in the collagen gel (e.g. in the absence and presence of high concentration of the compound(s)) would help the reader with data visualization/interpretation.

We added a figure of representative images of HBVSMC contraction in the collagen gel. Thank you

5) “Results”: Consider removing repetitive statements relevant to the use of statistical analysis in data interpretation for all Figures. Statistical analysis/approaches are well-described in the “Methods” section and in the Figure legends, as well.

We appreciate your attention to detail and have removed redundant language from the results regarding the statistical analysis. Thank you

6) Please label individually each panel in the Figures 1-4 and use this labeling in describing your findings in the “Results” section to help the reader with data interpretation.

We have added panel labels to the figures and referred to those subfigures in the results. We agree that it makes visualization and interpretation of the data clearer and appreciate the suggestion.

7) Figure 3. The data presented in Figure 3 (left panel) shows no difference in HBVSMC contraction in response to stimulation with 10-3M BDB on days 4 and 7, while the Figure 3 (center and right panels) indicate significant differences at these specific time points. Please clarify.

We appreciate your feedback and have worked to improve the reader’s ability to visualize the changes. Your previous suggestion of labeling the panels and referring to them in the text of the results has already helped. Additionally, we have included a line break in the Y axis of Panel A which helps to separate the data points to help with visualization of the differences. Thank you

8) Figure 6. The Figure/Diagram requires more detailed description/explanation of the potential effects/mechanisms involved.

We have added a more detailed description of Figure 6. Thank you

Reviewer 2 Report

I have reviewed the paper by Hall et al. and I believe the findings to be novel and worthy of publication.  I have a few minor comments that the authors can consider and that may improve the paper.

1. State the source of the HBVSMC

2. How were the drug concentrations determined. Were the concentrations used consistent with human plasma levels?

3. It was a model of chronic drug exposure. Were there any acute changes to be noted?  What is the half life of the drugs in human plasma? could the authors expand the discussion to include acute exposure versus chronic?

4. In the context of vasoconstriction, could the balance between constriction and dilation be discussed.  Would one expect NO levels to be altered?

5. Does reduced cellular contraction reflect inadequate vessel constriction?  If so, is this not contradictory to the drug-induced inadequate blood flow and ischemia observed in vivo?

6. The model requires more detail and explanation.

7. The term vascular homeostasis is used; perhaps more specific terminology would be helpful.

Author Response

  1. State the source of the HBVSMC

The source of the HBVSMC has been added. Thank you

  1. How were the drug concentrations determined. Were the concentrations used consistent with human plasma levels?

We have included information on how the drug concentrations were determined. Thank you

  1. It was a model of chronic drug exposure. Were there any acute changes to be noted?  What is the half life of the drugs in human plasma? could the authors expand the discussion to include acute exposure versus chronic?

We have expanded the discussion to include acute exposure. Thank you

  1. In the context of vasoconstriction, could the balance between constriction and dilation be discussed.  Would one expect NO levels to be altered?

Thank you for this comment. We have added the possibility of opposing vasodilation to the discussion.

  1. Does reduced cellular contraction reflect inadequate vessel constriction?  If so, is this not contradictory to the drug-induced inadequate blood flow and ischemia observed in vivo?

Although there is some evidence of drug-induced changes in cerebral blood flow localized ischemia, we do not feel there is adequate evidence to suggest an overall drug-induced inadequate blood flow and ischemia.

  1. The model requires more detail and explanation.

A more detailed explanation of Figure 6 has been added. Thank you

  1. The term vascular homeostasis is used; perhaps more specific terminology would be helpful.

The language has been corrected for clarity and references to vascular homeostasis were removed. Thank you

Reviewer 3 Report

In this study, HBVSMC suspended in collagen gels were expose to increasing methamphetamine or N-butylpentylone concentrations and analyzed for constriction/dilation or MTT. The authors stated that each drug produced direct effects on HBVSMC vascular homeostasis.

Serious flaws mine the entire manuscript from the experimental point of view. The correct evaluation of the contractile behavior of the cells is missing and the simple description of the three photographs (not even shown) is insufficient to establish if a real effect has been introduced in the cells.

The authors should provide pictures of this method as Supplementary material.

A correct data analysis should be carried out with a microscope and following the general guide for this kind of assays.

Moreover, in the figures 2-4 the central and right panels are already present in the left panel. All the figures should be collected in one, together with the MTT assays, as part A and B, respectively for the contracting and survival data in the presence of increasing compounds’ concentrations at different time points.

Altogether, results are insufficient to establish any of the aimed effects.

Finally, both the introduction and the discussion are long-winded and repetitive and lead to confusion and misleading.

Author Response

Serious flaws mine the entire manuscript from the experimental point of view. The correct evaluation of the contractile behavior of the cells is missing and the simple description of the three photographs (not even shown) is insufficient to establish if a real effect has been introduced in the cells.

The authors should provide pictures of this method as Supplementary material.

A correct data analysis should be carried out with a microscope and following the general guide for this kind of assays.

We appreciate your diligence in ensuring the integrity of the manuscript. We stand by this collagen gel contraction assay as an appropriate and proper method for the purposes of this manuscript. We and others have published this method before (PMID: 16799192, PMID: 34199925, PMID: 30257377, PMID: 35366270) and find it to be a useful and efficient assay to assess contraction of cells. We do, however, agree that visualization of the contractile behavior of the cells is necessary to fully appreciate the method. For this reason, we have added a figure to help the reader visualize how the method was carried out and analyzed.

Moreover, in the figures 2-4 the central and right panels are already present in the left panel. All the figures should be collected in one, together with the MTT assays, as part A and B, respectively for the contracting and survival data in the presence of increasing compounds’ concentrations at different time points.

 We do agree that the information in the center and right panels (now labeled B and C) are contained in the left panel (now labeled A); however, we feel it necessary to pull out select days to help the reader interpret the data as Panel A itself contains much more data than just the significant points. The clear separation of concentrations afforded by looking at one day at a time makes the data more digestible; although the big picture is important as well. Additionally, the MTT assay was not performed at different time points which could add confusion if combining the figures with the gel assay. Therefore, we feel it necessary to present the data as we have.

Round 2

Reviewer 3 Report

No real improvement has been done.

·         Correct cell sprouting quantification is missing;

·         Pictures’ improvement and reorganization are missing

·         Still, results are insufficient to establish any of the aimed effects

·         Introduction and Discussion show no improvements